

# Victory above all: the weight loss practices and perceptions of Chinese male kickboxers

Fanjie Meng[1], Yuming Zhong[2], Zhao Zhang[3] and Zihan Ren[4]

[1] Department of Martial Arts and Traditional National Sports, Henan Sport University, Henan, China
[2] School of Athletic Performance, Shanghai University of Sport, Shanghai, China
[3] Krirk University, Bangkok, Thailand
[4] Wushu College, Henan University, Henan, China

Corresponding author
Yuming Zhong,
2221111051@sus.edu.cn

## ABSTRACT

**Objectives.** Kickboxing is a dynamic combat sport (CS) in which two competitors engage in full-force strikes using their hands, elbows, knees, shins, and feet. To ensure athletes compete against opponents of similar body size and weight for fairness, kickboxing competition is classified according to weight classes. Previous studies revealed that CS athletes tend to intentionally lose weight to gain a competitive advantage over their opponents. However, little is known about weight loss (WL) practices in kickboxers and the perceptions of CS athletes about WL. The aim of this study is to investigate the WL practices and perceptions of Chinese male kickboxers.
**Methods.** A total of 152 Chinese male kickboxers completed the adapted Rapid WL Questionnaire. Participants were categorized by age group, weight category, competitive level, attitude toward impact of WL on health, and performance.
**Results.** Sixty-eight percent of kickboxers purposefully engaged in WL practices. The average habitual WL was 6.0% of body mass (BM), with the average highest WL being 8.8% of BM. Most participants reduced 41–60% of their total planned magnitude of WL between 60 days and 9 days before the weigh-in, 21–40% between 8 days and 2 days before the weigh-in, and 0–20% within 1 day before the weigh-in. Coaches (67%) were most reported as the primary guides for WL. The primary reason kickboxers engage in WL is competing against lighter opponents to increase the chances of winning (70%). Most participants believed that WL had no impact on health (42%), improved performance (57%), and did not lead to unfair competition (73%). Significant differences were observed in both the highest WL% ($p = 0.005$, $p = 0.018$), the habitual WL% ($p = 0.005$, $p = 0.018$), the age beginning WL ($p = 0.005$, $p = 0.018$), and the annual WL times ($p = 0.005$, $p = 0.018$) across age and competitive level, but not observed across weight category, and attitudes toward both the health and performance impacts of WL (all $p > 0.05$). No significant differences were found in habitual weight regain after weigh-in/habitual WL (%) after weigh-in across all groups (all $p > 0.05$).
**Conclusions.** The prevalence of WL among Chinese male kickboxers is relatively low when compared to other CS, but the magnitude is relatively higher compared with previous studies. Their WL practices are mainly guided by kickboxing coach, and the primary reasons of WL is to gain a competitive advantage and improve performance.

## BACKGROUND

Kickboxing is a dynamic combat sport (CS) in which two competitors engage in full-force strikes using their hands, elbows, knees, shins, and feet (*Slimani et al., 2017*). Kickboxing encompasses several distinct rulesets, such as low kick, K-1, and kick light. In most mainstream kickboxing rules, elbow strikes are generally prohibited, while knee strikes are permitted only under the K-1 ruleset. According to reports, World Association of Kickboxing Organizations (WAKO) has trained over 4 million practitioners in more than 40,000 clubs worldwide (*World Association of Kickboxing Organizations, 2022*). Additionally, kickboxing received provisional recognition from the International Olympic Committee in Tokyo, Japan, in 2018 and has been considered for inclusion in the 2028 Los Angeles Olympic Games (*World Association of Kickboxing Organizations, 2022*). These developments reflect the sport's growing international prominence. Like other CS (such as boxing and MMA), kickboxing matches are held in weight classes to ensure fairness. Athletes who fail to meet the required weight may, depending on the specific competition rules and organizer discretion, be disqualified or allowed to compete in a higher weight category, often with associated penalties or conditions.

Past studies have found that most CS athletes lose weight before competition to gain a competitive advantage over their opponents (*e.g.*, fighting an opponent with smaller physicality) (*Berkovich et al., 2016*; *Brito et al., 2012*; *Dugonjić, Krstulović & Kuvačić, 2019*; *Giannini Artioli et al., 2010*). More specifically, practitioners lose weight prior to weigh-in to fall into a particular weight category, then make efforts to gain as much weight between weigh-in and the fight. This is believed to provide a perceived competitive advantage (*Pettersson, Ekström & Berg, 2013*). Despite evidence that rapid WL can negatively affect athletes' health and performance (*Andreato et al., 2025*; *Artioli et al., 2016*; *Degoutte et al., 2006*; *Franchini, Brito & Artioli, 2012*; *Gann, Tinsley & La Bounty, 2015*; *Pélissier et al., 2023*; *Štangar et al., 2022*; *Vasconcelos, Guedes & Del Vecchio, 2023*; *Yoshida et al., 2024*), a gap remains between research findings and real-life practice, as many athletes continue to employ WL strategies (*Zhong et al., 2024b*).

Extensive research compiled in a systematic review has investigated the WL practices of CS athletes from different sports such as judo, mixed-martial arts (MMA), and sambo, across different countries (*e.g.*, Brazil, France, and Australia) and all competitive levels (*e.g.*, high school, university, and elite) (*Zhong et al., 2024b*). This systematic review found similarities and differences in the WL practices of CS athletes from different backgrounds (sport, country, and competitive level). Similarities include CS athletes (i) commonly adopting WL practices (66–100%) irrespective of the athlete's demographic background; (ii) using increased exercise and gradual dieting as the most common methods for WL; and (iii) losing < 5% of BM 7–14 days before a competition (*Zhong et al., 2024b*). However, significant variations in the WL practices (such as habitual WL percentage of BM (WL%)) have also been observed, with national and sport-specific differences. For example, a difference in habitual WL% can be seen between MMA athletes in Brazil (13%) and Ireland (7.9%) (*Connor & Egan, 2019*; *Santos-Junior et al., 2020*), and between kickboxing (4.5%) and MMA (13%) athletes in Brazil (*Santos-Junior et al., 2020*; *Vasconcelos, Guedes*

& *Del Vecchio, 2023*). A recent systematic review also identified the sport-specific nature of WL practices among CS athletes, with both habitual and highest WL% of MMA, Muay Thai, and sambo athletes being significantly higher than those of athletes in other sports (*e.g.*, boxing, kickboxing, and judo) (*Zhong et al., 2024b*). The unique WL practice may be related to the unique culture and specific competition rules of the sports. Aggressive WL in MMA, Muay Thai, and sambo athletes could stem from (i) single weigh-in prior to competition; (ii) weigh-in usually conducted 24–36 h prior to competition; and (iii) a culture of aggressive WL being entrenched and accepted within these communities (*Park et al., 2019*; *Santos-Junior et al., 2020*).

Although extensive past research has provided valuable insights into the WL practices of CS athletes (*Baranauskas, Kupčiūnaitė & Stukas, 2022*; *Castor-Praga, Lopez-Walle & Sanchez-Lopez, 2021*; *Drid et al., 2021*; *Figlioli et al., 2021*; *Kons et al., 2022*; *Nolan, Lynch & Egan, 2022*; *Pélissier et al., 2023*; *Ranisavljev et al., 2022*; *Roklicer et al., 2022*; *Vasconcelos, Guedes & Del Vecchio, 2023*; *White & Kirk, 2021*), only one study has investigated kickboxers (*Dugonjić, Krstulović & Kuvačić, 2019*), and it was limited to the elite level, which significantly hinders the understanding of WL practices among kickboxers. In K1 rule kickboxing, if an athlete fails the weigh-in, they are still allowed a second weigh-in opportunity (*World Association of Kickboxing Organizations, 2020*). This unique rule may lead to kickboxers employing different WL practices compared to athletes in other CS, making this an area worth exploring. In addition, although previous studies have explored the WL practices of CS athletes, only few researches have investigated their perceptions of WL (*Pettersson, Ekström & Berg, 2013*; *Smith et al., 2024*; *Tako et al., 2020*), which is a significant gap in the current literature. Understanding athletes' perceptions of WL may help explain why certain harmful practices are adopted despite known risks. Therefore, another aim of this study is to investigate the perceptions of WL among kickboxers.

Given the current lack of understanding regarding both the practices and perceptions of WL among kickboxers, an investigation into the practices and perceptions of WL among Chinese kickboxers would provide valuable insights into this understudied population. Thus, the objective of this research is to investigate the practices and perceptions of WL among Chinese male kickboxers, using an adapted version of the Rapid Weight Loss Questionnaire (RWLQ) to provide detailed and contemporary insights into this understudied population.

## MATERIALS & METHODS

### Experimental approach to the problem

The study employed an observational cross-sectional approach using a questionnaire to ascertain the practices and perceptions of Chinese male kickboxers. To target the relevant population, this study adopted a mixed non-probability sampling approach, combining purposive sampling and convenience sampling. The study followed the cross-sectional reporting guidelines outlined by the Strengthening the Reporting of Observational Studies in Epidemiology (STROBE) (*Von Elm et al., 2014*), ensuring transparency and consistency in observational research practices.

## Participants

Participants were recruited both online and offline. For online recruitment, a questionnaire link was sent to the managers of kickboxing sports teams through social media (WeChat, Tencent, China), requesting this to be shared with their athletes (*Zhong et al., 2023*). Paper questionnaires were distributed offline at the largest training base for CS athletes in China. The kickboxing team managers assisted in recruiting participants during daily sports team meetings. All athletes who agreed to participate in the offline questionnaire were gathered in the conference room to complete the questionnaire (*Drid et al., 2021*). If any questions were unclear, researchers were available to provide detailed explanations. For online questionnaires, some simple answering prompts are added in the questionnaires. Before the application of the questionnaire, participants were orally briefed on the instrument for the offline survey, while written information was provided for the online survey before signing informed consent. For junior athletes, informed assent was obtained from the participants, and informed consent was obtained from their parents or guardians before the questionnaire session. The inclusion criteria for participants were: (i) consent to participate in research, (ii) participation in at least one official kickboxing competition in the last 12 months, (iii) currently training as an athlete on a sports team, (iv) does not compete in the 100+kg division, and (v) male athlete. In total, 193 participants completed the questionnaire. The lead author (FJM) pre-checked all responses, and 20 responses were considered invalid, of which 19 participants were excluded from criterion (ii) and two participants from criterion (iv). Finally, 152 responses from male kickboxers were included in the analysis. Participants were categorized by age (junior < 18 years old, senior ≥ 18 years old), competition weight categories (lightweight −51 kg to −63.5 kg, middleweight −67 kg to 75 kg, heavyweight −81 kg to −91 kg) (*Dugonjić, Krstulović & Kuvačić, 2019*), competitive levels (local, provincial, national, international), attitude toward impact of WL on health (no impact, detrimental to health, beneficial to health), and performance (no impact, detrimental to performance, beneficial to performance). Approval for conducting the study was secured from the Ethics Committee of the Shanghai University of Sport (number of approval: 102772023RT170).

## Questionnaire development and administration

Open access application WJX web platform (Wenjuanxing, http://www.wjx.cn) was used to construct, distribute, and collect all questionnaire responses. Responses from paper questionnaires were manually input into the WJX web platform by the lead author (FJM). This study employed the pre-validated RWLQ by *Artioli et al. (2010)*, which has been used in previous studies on WL practices in Chinese boxers (*Zhong et al., 2024a*). The RWLQ is widely used for CS (*Zhong et al., 2024b*) and non-CS athletes (*Alwan et al., 2022*; *Gee et al., 2023*; *Meroño, Frideres & Palao, 2019*). This study added several questions to the original RWLQ to further explore athletes' WL practices and perceptions. The additional questions include the following: (1) the proportion of total BM lost during different stages before competition, (2) who guides the WL process, (3) the habitual weight regain (WR) after weigh-in, (4) the reasons for WL, (5) the impact of WL on health, (6) the impact of WL on performance, and (7) whether WL leads to unfairness in competitions. The

questionnaire consisted of 30 questions and featured four sections: (i) general information, (ii) competition experience, (iii) weight history, and (iv) WL perception. The content validity of the Chinese version of the questionnaire was evaluated by a researcher with extensive experience in the field of kickboxing and five kickboxers through pilot tests. The researchers helped check the wording of the questionnaire to make sure it was correct, and the athletes completed it to try to spot anything they could not understand or that they thought should be adjusted. Pilot testing led to minor modifications in the wording of certain questions to ensure clarity and relevance for kickboxers. Data collection was conducted from November 15, 2024 to January 1, 2025.

## Statistical analyses

All analyses were performed using SPSS 27.0 (IBM Corp., Armonk, NY, USA). Descriptive statistics were used to summarize all results. Continuous variables were presented as mean ($\pm$ SD), and categorical variables as frequencies (%). The Kolmogorov–Smirnov test was used to check data normality. All continuous variables, except for the BM and off-season BM, violated parametric assumptions of normality. Independent $t$-tests were used to analyze differences in BM and off-season BM between participants who engaged in WL and those who never engaged in WL. Mann–Whitney U tests were used to analyze differences in age, stature, age began training in kickboxing, age began competing in kickboxing, competitions participated in last competitive season, medals gained during competitive season between participants who engaged in WL and those who never engaged in WL. Mann–Whitney U tests were also used to analyze differences in highest WL%, habitual WL%, habitual WR after weigh-ins/habitual WL (%), age began WL, and number of WL in the last year between age groups. Kruskal–Wallis' tests were used to analyze differences in highest WL%, habitual WL%, habitual WR after weigh-ins/habitual WL (%), age began WL, and number of WL in the last year across competitive levels and weight categories, attitudes toward health impacts of WL, and attitudes toward performance impacts of WL. Significance was accepted at $p < 0.05$. To address the issue of inflated significance levels due to multiple comparisons, Bonferroni adjustments were applied for $p$ value using $R$ (version 4.3.0; R Core Team, Vienna, Austria). In this study, only the general information, highest WL%, habitual WL%, number of WL in the last year, habitual WR after weigh-ins/habitual WL (%), and age began WL were analyzed using inferential statistics, as these variables are closely related to our main research question and exhibit a logical relationship. In contrast, inferential statistics was not performed on some other variables due to a lack of correlation among the variables (such as differences in age began WL between age group), making testing them meaningless. The calculation formula for habitual WR after weigh-ins/habitual WL (%) was "habitual WR after weigh-ins (kg)/habitual WL (kg)". The frequencies of responses indicating 'always' and 'sometimes' for WL methods were combined to describe the participants' primary WL methods, as both represent current use. The frequency of choice of "very influential" and "influential" in sources of influence is combined to describe the sources that have influence for athletes on their WL practices, as both choices represent influential and others represent no clear influence.

## RESULTS

One hundred and fifty-two male kickboxers participated in this study. Participants' general information is presented in Table 1. Significant differences were only observed in competitions participated last competitive season ($p = 0.007$) between those who engaged in WL and those who did not. In contrast, no significant differences were found in age ($p = 1.000$), stature ($p = 0.464$), BM ($p = 0.328$), age began training in kickboxing ($p = 1.000$), age began competing in kickboxing ($p = 1.000$), and medals gained during last competitive season ($p = 0.392$) and off-season BM ($p = 0.608$) between participants who engaged in WL and those who did not. In total, 68% of participants ($n = 104$) reported purposefully performed WL for competition, and 63% had not changed their weight category in the past two years.

Only those who engaged in WL ($n = 104$) answered the following questions. The results of the participants' WL history are shown in Table 2. Significant differences were observed in both the highest WL% ($p = 0.005$, $p = 0.018$), the habitual WL% ($p = 0.005$, $p = 0.018$), the age began WL ($p = 0.005$, $p = 0.018$), and the annual WL times ($p = 0.005$, $p = 0.018$) across age and competitive level, but not observed across weight category, and attitudes toward both the health and performance impacts of WL (all $p > 0.05$). No significant differences were found in habitual WR after weigh-ins/habitual WL (%) across all groups (all $p > 0.05$). Participants usually allocated 15+ days before the competition for WL (43%), followed by 11–14 days (18%), 8–10 days (16%), 4–5 days (13%), 6–7 days (6%), and 1–3 days (4%). Most participants reduced 41–60% of their total planned magnitude of WL between 60 days and 9 days before the weigh-in, 21–40% between 8 days and 2 days before the weigh-in, and 0–20% within 1 day before the weigh-in. Coach (67%) were most commonly reported as the primary guides for WL, followed by self-guidance (57%), strength and conditioning coach (4%), parent (2%), doctor (1%) and nutritionist (0%). The primary reason athletes engage in WL is competing against lighter opponents to increase the chances of winning (70%), followed by optimizing athletic performance (53%), being above my usual weight before competition (39%), and because everyone else is cutting weight, so I have to do it too (8%). Most participants believed that WL had no impact on health (42%), followed by those who thought it is detrimental to health (39%) or beneficial to health (19%). Most participants believed that WL is beneficial to performance (57%), followed by those who thought it is detrimental to performance (28%) or had no impact (15%). The majority of participants believed that WL did not lead to unfair competition (73%), followed by those who were uncertain (26%) or believed it did result in unfair competition (1%). The WL methods used by athletes are shown in Table 3. The influence of different sources on the WL practices reported by athletes on their WL is shown in Table 4.

## DISCUSSION

This study investigated the WL practices and perceptions of Chinese male kickboxers. The primary findings include: (i) the prevalence (68%) of WL is relatively low when compared to other CS, but the habitual WL% and highest WL% was relatively high when compared

Meng et al. (2025), *PeerJ*, DOI 10.7717/peerj.19709

**Table 1  General information (mean and standard deviation) of Chinese male kickboxers (*n* = 152).**

| Variables | All participants (*n* = 152) | | | Participants who never engaged in weight loss (*n* = 48) | | | Participants who engaged in weight loss (*n* = 104) | | | *P* |
|---|---|---|---|---|---|---|---|---|---|---|
| | Mean ± SD | Max | Min | Mean ± SD | Max | Min | Mean ± SD | Max | Min | |
| Age (years) | 17.2 ± 2.8 | 29 | 12 | 16.5 ± 1.6 | 19 | 13 | 17.4 ± 3.2 | 29 | 12 | 1.000 |
| Body mass (kg) | 65.3 ± 12.2 | 106.0 | 35.0 | 62.4 ± 12.7 | 106.0 | 39.0 | 66.7 ± 11.7 | 90.0 | 35.0 | 0.328 |
| Stature (cm) | 174.6 ± 7.6 | 193.0 | 147.0 | 173.1 ± 8.0 | 187.0 | 155.0 | 175.2 ± 7.4 | 193.0 | 147.0 | 0.464 |
| Age began training in kickboxing (years) | 13.8 ± 2.2 | 22 | 8 | 14.0 ± 1.9 | 18 | 9 | 13.7 ± 2.3 | 22 | 8 | 1.000 |
| Age began competing in kickboxing (years) | 15.0 ± 2.2 | 22 | 9 | 15.0 ± 1.5 | 18 | 13 | 15.1 ± 2.5 | 22 | 9 | 1.000 |
| Off-season body mass (kg) | 65.0 ± 12.6 | 106.0 | 35.0 | 62.4 ± 13.3 | 106.0 | 35.0 | 66.3 ± 12.1 | 91.0 | 35.0 | 0.608 |
| Competitions participated last competitive season (*n*) | 2.2 ± 2.5 | 15 | 1 | 1.2 ± 0.6 | 4 | 1 | 2.7 ± 2.9 | 15 | 1 | 0.007[*] |
| Medals gained during last competitive season (*n*) | 1.6 ± 1.7 | 8 | 0 | 1.1 ± 1.0 | 4 | 0 | 1.8 ± 2.0 | 8 | 0 | 0.392 |

**Notes.**
*Differences between participants who engaged in weight loss and those who never engaged in weight loss (*P* < 0.05). SD, standard deviation; Max, maximum; Min, minimum.

Meng et al. (2025), *PeerJ*, DOI 10.7717/peerj.19709

**Table 2  Weight loss history (mean and standard deviation) of Chinese male kickboxers who reported engaging in weight loss practices ($n = 104$).**

| Variables | Total ($n = 104$) | Junior ($n = 64$) | Senior ($n = 40$) | P | Local ($n = 71$) | Provincial ($n = 10$) | National ($n = 15$) | International ($n = 8$) | P | Lightweight ($n = 59$) | Middleweight ($n = 33$) | Heavyweight ($n = 12$) | P |
|---|---|---|---|---|---|---|---|---|---|---|---|---|---|
| Age began WL (yr) | 15.4 ± 2.2 | 14.3 ± 1.6 | 17.1 ± 1.7 | 0.005[*] | 14.7 ± 1.8 | 16.7 ± 2.3 | 16.5 ± 1.7 | 17.9 ± 2.3 | 0.018[*] | 14.9 ± 2.2 | 15.9 ± 1.9 | 16.2 ± 2.0 | 0.240 |
| Highest WL (kg) | 6.0 ± 3.1 | 4.8 ± 2.6 | 7.9 ± 2.8 | / | 5.0 ± 2.7 | 7.6 ± 3.2 | 7.5 ± 2.9 | 9.5 ± 1.9 | / | 4.8 ± 2.2 | 7.3 ± 3.5 | 8.2 ± 3.1 | / |
| Highest WL (%) | 8.8 ± 3.8 | 7.6 ± 3.6 | 10.8 ± 3.3 | 0.005[*] | 7.9 ± 3.7 | 10.0 ± 3.1 | 10.7 ± 3.3 | 12.7 ± 2.3 | 0.018[*] | 8.1 ± 3.3 | 9.9 ± 4.5 | 9.6 ± 3.5 | 1.000 |
| Habitual WL (kg) | 4.1 ± 2.3 | 3.3 ± 2.0 | 5.3 ± 2.3 | / | 3.3 ± 1.9 | 5.6 ± 2.7 | 5.0 ± 2.1 | 7.3 ± 1.7 | / | 3.2 ± 1.7 | 5.2 ± 2.4 | 5.3 ± 2.9 | / |
| Habitual WL (%) | 6.0 ± 2.9 | 5.2 ± 2.6 | 7.3 ± 2.8 | 0.005[*] | 5.2 ± 2.7 | 7.3 ± 2.7 | 7.0 ± 2.3 | 9.8 ± 2.2 | 0.018[*] | 5.4 ± 2.6 | 7.0 ± 3.0 | 6.2 ± 3.2 | 1.000 |
| Habitual WR (kg) | 2.9 ± 2.1 | 2.1 ± 1.6 | 4.0 ± 2.2 | / | 2.2 ± 1.8 | 4.0 ± 1.9 | 3.8 ± 2.2 | 5.1 ± 2.1 | / | 2.2 ± 1.5 | 3.7 ± 2.4 | 3.8 ± 2.3 | / |
| WR/WL (%) | 74.6 ± 48.8 | 68.1 ± 47.0 | 85.0 ± 50.3 | 0.385 | 73.0 ± 54.0 | 77.0 ± 19.8 | 80.7 ± 44.5 | 74.1 ± 34.8 | 1.000 | 68.8 ± 42.1 | 79.0 ± 50.7 | 90.9 ± 69.9 | 1.000 |
| Number of WL in the last year | 2.5 ± 2.6 | 1.8 ± 1.4 | 3.6 ± 3.5 | 0.005[*] | 2.0 ± 2.3 | 5.1 ± 3.1 | 2.0 ± 1.4 | 4.5 ± 3.0 | 0.018[*] | 1.9 ± 1.5 | 3.5 ± 3.8 | 2.2 ± 1.5 | 1.000 |

**Notes.**

*Differences between groups ($P < 0.05$). WL, weight loss. WR, weight regain. WR/WL (%), Habitual WR after weigh-ins/habitual WL (%).

**Table 3** Frequency analysis (%) of the weight loss methods used by Chinese male kickboxers (n = 104).

| | Always | | Sometimes | | Almost never | | Never used | | Do not use any more | |
|---|---|---|---|---|---|---|---|---|---|---|
| | *n* | % | *n* | % | *n* | % | *n* | % | *n* | % |
| Gradual dieting | 30 | 20 | 25 | 16 | 17 | 11 | 27 | 18 | 5 | 3 |
| Skipping meals | 13 | 9 | 44 | 29 | 16 | 11 | 21 | 14 | 10 | 7 |
| Fasting | 12 | 8 | 14 | 9 | 19 | 13 | 51 | 34 | 8 | 5 |
| Restricting fluid ingestion | 22 | 14 | 28 | 18 | 16 | 11 | 30 | 20 | 8 | 5 |
| Increased exercise | 49 | 32 | 31 | 20 | 7 | 5 | 15 | 10 | 2 | 1 |
| Training in a heated room | 34 | 22 | 26 | 17 | 17 | 11 | 22 | 14 | 5 | 3 |
| Sauna | 13 | 9 | 24 | 16 | 18 | 12 | 45 | 30 | 4 | 3 |
| Training in plastic suits | 39 | 26 | 32 | 21 | 12 | 8 | 16 | 11 | 5 | 3 |
| Use plastic suit all-day | 8 | 5 | 13 | 9 | 21 | 14 | 57 | 38 | 5 | 3 |
| Spitting | 6 | 4 | 12 | 8 | 18 | 12 | 62 | 41 | 6 | 4 |
| Laxatives | 1 | 1 | 3 | 2 | 8 | 5 | 86 | 57 | 6 | 4 |
| Diuretics | 1 | 1 | 3 | 2 | 13 | 9 | 80 | 53 | 7 | 5 |
| Diet pills | 1 | 1 | 3 | 2 | 7 | 5 | 87 | 57 | 6 | 4 |
| Vomiting | 2 | 1 | 8 | 5 | 12 | 8 | 76 | 50 | 6 | 4 |
| Hot water immersion | 5 | 3 | 17 | 11 | 9 | 6 | 67 | 44 | 6 | 4 |
| Hot salt water immersion | 2 | 1 | 6 | 4 | 9 | 6 | 80 | 53 | 7 | 5 |
| Others | 4 | 3 | 5 | 3 | 9 | 6 | 78 | 51 | 8 | 5 |

**Table 4** Frequency analysis (%) of the sources of influence on the weight loss practices of Chinese male kickboxers (n = 104).

| | Very influential | | Some influence | | Unsure | | Little influence | | Not influential | |
|---|---|---|---|---|---|---|---|---|---|---|
| | *n* | % | *n* | % | *n* | % | *n* | % | *n* | % |
| Other athletes (different sports) | 3 | 2 | 16 | 11 | 17 | 11 | 17 | 11 | 51 | 34 |
| Other athletes (same sport) | 13 | 9 | 24 | 16 | 14 | 9 | 26 | 17 | 27 | 18 |
| Doctors | 3 | 2 | 11 | 7 | 21 | 14 | 20 | 13 | 49 | 32 |
| Strength and conditioning coaches/ physical trainer | 8 | 5 | 26 | 17 | 21 | 14 | 8 | 5 | 41 | 27 |
| Coaches | 26 | 17 | 32 | 21 | 12 | 8 | 8 | 5 | 26 | 17 |
| Parents | 15 | 10 | 15 | 10 | 18 | 12 | 11 | 7 | 45 | 30 |
| Nutritionists | 5 | 3 | 8 | 5 | 19 | 13 | 11 | 7 | 61 | 40 |
| Journal articles | 1 | 1 | 7 | 5 | 17 | 11 | 18 | 12 | 61 | 40 |
| Book/magazines | 1 | 1 | 12 | 8 | 18 | 12 | 17 | 11 | 56 | 37 |
| Internet sources | 3 | 2 | 20 | 13 | 18 | 12 | 14 | 9 | 49 | 32 |
| Others | 2 | 1 | 7 | 5 | 18 | 12 | 10 | 7 | 67 | 44 |

to other CS; (ii) significant differences in most WL practices (habitual WL%, highest WL%, age began WL, and annual WL times) were only observed among age groups and competitive levels, but not across weight categories, attitudes toward the performance impact of WL, and attitudes toward the health impact of WL; (iii) Most participants reduced 41–60% of BM between 60 days and 9 days before the weigh-in, 21–40% of BM between 8 days and 2 days before the weigh-in, and 0–20% of BM within 1 day before the weigh-in; (iv) Most participants believed that WL had no impact on health, improved

performance, and did not lead to unfair competition; and (v) coaches, fellow kickboxers, and parents were influential in the WL practices employed.

In this study, 68% of kickboxers purposefully engaged in WL practices, which is lower than most previous findings from other CS and countries (*Zhong et al., 2024b*). Similarly, kickboxers in this study engaged in WL practices to a lesser prevalence than elite kickboxers from 8 European countries (100%) (*Dugonjić, Krstulović & Kuvačić, 2019*). The relative lower prevalence of WL in this study may be due to the fact that most participants were local-level and junior kickboxers, groups typically exhibiting lower WL prevalence (*Giannini Artioli et al., 2010*; *Zhong et al., 2024a*; *Zhong et al., 2024b*). Additionally, in this study, individuals who engaged in WL had a significantly higher annual competition frequency compared to those who did not, suggesting that an increase in competition frequency significantly enhances the likelihood of engaging in WL.

The average habitual WL of participants was 6.0% of BM (Table 2), higher or similar to most previous studies in different sports (*Zhong et al., 2024b*), and only lower than results from MMA (6.7–13.0%) (*Andreato et al., 2014*; *Barley, Chapman & Abbiss, 2018*; *Connor & Egan, 2019*; *Hillier et al., 2019*; *Ribas et al., 2017*; *Santos-Junior et al., 2020*) and Muay Thai (10.6%) (*Ribas et al., 2019*). In this study, the average highest WL of athletes was 8.8% of BM, higher than results from judo (4.7%−6.0%) (*Berkovich et al., 2016*; *Giannini Artioli et al., 2010*), taekwondo (7.3%) (*Da Silva Santos et al., 2016*), and boxing in Italy (5.5%) (*Amatori et al., 2020*), but similar to kickboxing (8.4%) (*Dugonjić, Krstulović & Kuvačić, 2019*) and multiple CS studies (8.0−8.5%), and lower than results from than results from boxing in China (9.5%) (*Zhong et al., 2024a*), visually impaired judo (10.0%) (*Kons et al., 2022*), sambo (10.6%) (*Figlioli et al., 2021*), Sanda (10.3%) (*Vasconcelos, Guedes & Del Vecchio, 2023*), MMA (10.2%–17.5%) (*Andreato et al., 2014*; *Ribas et al., 2017*; *Santos-Junior et al., 2020*), Muay Thai (13.9%) (*Ribas et al., 2019*). These results indicate that the habitual WL% of the kickboxers in present study was high, only lower than MMA athletes. However, the competition rule of MMA provides sufficient time (generally between 24 and 36 hr) for athletes to recover and athletes only need to be weighed once (*Coswig et al., 2019*; *Matthews & Nicholas, 2017*). In contrast, Chinese kickboxing competitions require athletes to weigh in on the morning of each competition day (*Hunan Daily, 2023*), leaving a short recovery window that makes it challenging for athletes to recover from a weakened state.

Compared to other sports (such as boxing and Sanda) with the same weighing rules, the WL% of these kickboxers is still higher, as the above comparison, highlighting the severe WL practices of Chinese kickboxers. However, Chinese kickboxers allocated a longer time for WL process, which can reduce the negative impact of high-volume WL to some extent. In present study, senior kickboxers demonstrated higher highest WL% (10.8% *vs* 7.6%) and habitual WL% (7.3% *vs* 5.2%) compared to junior kickboxers. This could be due to their greater experience with WL and more mature physical development, enabling them to tolerate higher WL magnitude. Additionally, international-level athletes showed significantly higher WL% than athletes at other competitive levels. This suggests that higher levels of competition drive athletes to seek a competitive edge through WL, even though WL $\geq$ 5% BM can negatively affect physical and mental health (*Franchini, Brito & Artioli,*

2012; *Mauricio et al., 2022*). On average, these kickboxers regained 74.6% of their reduced BM between weigh-ins and competitions; however, some athletes regained 250% (lost two kg, regain five kg) of the reduced BM, likely to gain a competitive advantage. WR has been associated with success in MMA and judo competitions (*Coswig et al., 2019*; *Reale et al., 2016*), suggesting that some athletes are manipulating WR for a performance boost. The primary reason athletes engaged in WL was to compete against lighter opponents to increase their chances of winning, which helps explain the high WL% and WR%.

Regarding WL duration, 43% of participants allocated 15+ days before competition for WL, which is higher than in other studies that reported shorter WL durations (*Giannini Artioli et al., 2010*; *Kons et al., 2017*; *White & Kirk, 2021*; *Yarar, Turkyilmaz & Eroglu, 2019*). Specifically, most participants reduced 41–60% of their total planned magnitude of WL between 60 days and 9 days before the weigh-in, 21–40% between 8 days and 2 days before the weigh-in, and 0–20% within 1 day before the weigh-in. This is expected because the WL% of these athletes is relatively high, and longer durations are required to mitigate the associated adverse effects of RWL. Previous studies in wrestling has shown that gradual WL before competition did not affect peak torque of knee flexion and extension movements, support the reduction of body fat, and increase lean body mass (*Miranda et al., 2021*). Similarly, a study on male CS athlete found that 7 weeks of daily energy availability (EA) fluctuations averaging 20 kcal kg fat-free mass $(\text{FFM})^{-1}$ $\text{d}^{-1}$ led to reductions in BM and FM without disrupting physiological systems related to the Male Athlete Triad and Relative Energy Deficiency in Sport (RED-S). In contrast, a subsequent period of 5 consecutive days of $\text{EA} < 10 \, \text{kcal} \, \text{kg} \, \text{FFM}^{-1} \, \text{d}^{-1}$ induced consequences associated with the Male Athlete Triad and RED-S (*Langan-Evans et al., 2021*).

The most frequently used WL methods by participants were increasing exercise (52%), training in plastic suits (47%), skipping meals (38%), training in heated rooms (38%), gradually dieting (36%), and restricting fluid intake (32%) (Table 3). This finding is consistent with most previous studies (*Zhong et al., 2024b*). This is possibly due to these methods being more accessible and easier to implement than alternatives (*e.g.*, advanced equipment or medication), and are also more likely to be recommended by the sources of influence that are most important to them (kickboxing coach). However, while training in plastic suits was acknowledged as a prevalent WL method in other studies (*Drid et al., 2021*; *Dugonjić, Krstulović & Kuvačić, 2019*; *Giannini Artioli et al., 2010*; *Kons et al., 2017*; *Malliaropoulos et al., 2018*; *Todorović et al., 2021*; *White & Kirk, 2021*), it is typically regarded as a supplementary, often ranking between fifth and seventh in usage frequency. In contrast, its primary ranking (second) in our findings suggests a distinctive reliance on this method among kickboxers. This significant difference may be attributed to several factors. First, in Chinese kickboxing teams, a single coach is typically responsible for 20–40 athletes, there is little effort devoted to developing personalized WL strategies, making the use of easily accessible plastic suits the second most practical choice. Second, athletes' WL methods are primarily guided by their coaches, leading to a greater reliance on training-based approaches (increasing exercise and training in plastic suits). Third, wearing plastic suits can effectively lose BM in daily training (*Park et al., 2025*), so athletes do not need to carry out additional WL activities. Similarly, previous research on WL

practices among Chinese boxers also identified athletes' preference for plastic suits (*Zhong et al., 2024a*). It is worth noting that some harmful methods (fasting, laxatives, diuretics, diet pills, vomiting) are also used, albeit less frequently (Table 3).

Consistent with previous findings (*Zhong et al., 2024b*), sports coach (38%), other kickboxer (25.0%), S&C coaches/physical trainers (22%), and parents (20%) were considered to have the highest impact on athletes' WL practices (Table 4). Similarly, previous investigations on Chinese boxers have also reported comparable patterns (*Zhong et al., 2024a*). Notably, although these roles remain the most influential, the values reported in this study are considerably lower than those observed in Chinese boxers, suggesting a weaker influence of these roles on kickboxers, possibly due to the different organizational structure and lower competitive levels.

Findings from this study indicate that coaches are typically the primary figures guiding athletes in WL practices, followed by self-guidance. This is particularly concerning, as coaches often lack adequate knowledge of nutrition and physiology, making them more likely to pass down personal experiences accumulated during their own athletic careers rather than evidence-based strategies (*Berkovich et al., 2019*; *Danaher & Curley, 2014*; *Sung, Lee & Lee, 2024*). Similarly, athletes who rely on self-guided WL practices may adopt harmful methods, especially when coaches are unavailable or lack expertise. It is worth noting that only one athlete received a doctor's guidance during the usual WL process, and none had a nutritionist's guidance. The absence and low influence of nutritionists and doctors in the WL process is a common phenomenon, largely because the primary goal of both coaches and athletes is to win the competitions.

In this study, although 40% of participants believed that WL negatively impacts health, they still continued their WL practices because they believed that WL could optimize performance and allow them to compete against lighter opponents. Additionally, a small portion (19%) of athletes even believed that WL could improve health. A lack of basic nutritional knowledge may lead athletes to adopt more aggressive WL practices, as most of them aim to compete against lighter opponents and optimize their performance. Therefore, sports team managers should prioritize research on WL practices and the dissemination of accurate knowledge, while also strengthening collaboration among coaches, nutritionists/medical professionals, athletes, and parents. By leveraging the influence of coaches and parents alongside the expertise of nutritionists/doctors, WL strategies can be optimized to ensure both safety and effectiveness for athletes. Therefore, we recommend that WL practices in kickboxing (and other CS) be guided by certified professionals such as sports dietitians and physicians, rather than relying solely on coaches or self-guidance. Specific methods such as training in plastic suits and fasting should be discouraged due to their potential health risks and lack of long-term efficacy. Instead, gradual and evidence-based WL strategies should be promoted. Athletic organizations should develop clear guidelines and educational initiatives to ensure athletes make informed and safe WL decisions. These steps are crucial not only for performance optimization but also for the long-term well-being of CS athletes.

This study has two main limitations. First, it relied on self-reported data, which may introduce biases, including recall and social desirability bias. To improve future research,

more objective data collection methods should be considered. Second, the sample size imbalance across competitive level is another limitation, as it may be underpowered to detect significant differences in WL history variables across competitive levels.

## CONCLUSION

Compared to previous studies, the prevalence of WL is relatively low amongst Chinese male kickboxers, but the magnitudes are very high when compared to other CS. Most athletes usually allocated a longer time to lose a high percentage of their BM compared to other CS athletes. Coach was most commonly reported as the primary guides for WL. The primary reason male kickboxers engage in WL is competing against lighter opponents to increase the chances of winning. Most participants believed that WL had no impact on health, improved performance, and did not lead to unfair competition. Increasing exercise was the most frequently used WL methods and kickboxing coach was considered to have the highest influence on athletes' WL practices. Professional nutrition support does not seem to receive due attention in Chinese kickboxing teams.

## ACKNOWLEDGEMENTS

The authors thank the participants for the time spent completing the questionnaire.

### Funding

This work was supported by the 2023 Henan Provincial Department of Science and Technology Key Research Project (No. 232102321126): Research on the Protection and Inheritance of Henan Martial Arts Cultural Resources Based on Information Technology. The funders had no role in study design, data collection and analysis, decision to publish, or preparation of the manuscript.

### Grant Disclosures

The following grant information was disclosed by the authors:
The 2023 Henan Provincial Department of Science and Technology Key Research Project: Research on the Protection and Inheritance of Henan Martial Arts Cultural Resources Based on Information Technology: No. 232102321126.

### Competing Interests

The authors declare there are no competing interests.

### Author Contributions

- Fanjie Meng conceived and designed the experiments, performed the experiments, analyzed the data, prepared figures and/or tables, authored or reviewed drafts of the article, and approved the final draft.
- Yuming Zhong conceived and designed the experiments, performed the experiments, analyzed the data, prepared figures and/or tables, authored or reviewed drafts of the article, and approved the final draft.

- Zhao Zhang performed the experiments, authored or reviewed drafts of the article, and approved the final draft.
- Zihan Ren performed the experiments, authored or reviewed drafts of the article, and approved the final draft.

## Human Ethics

The following information was supplied relating to ethical approvals (i.e., approving body and any reference numbers):

Approval for conducting the study was secured from the Ethics Committee of the Shanghai University of Sport (102772023RT170).

## Data Availability

The raw data and code are available in the Supplementary Files.

## Supplemental Information

Supplemental information for this article can be found online at http://dx.doi.org/10.7717/peerj.19709#supplemental-information.

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
