# Peer review of "Victory above all: the weight loss practices and perceptions of Chinese male kickboxers"

_PeerJ, doi:10.7717/peerj.19709_

## Round 0.1 · original submission · Major Revisions

· Academic Editor

Major Revisions

Please see the attached reviews to improve your manuscript.

·

Basic reporting

Lines 52-53
Kickboxing is a dynamic combat sport (CS) in which two competitors engage in full-force strikes using their hands, elbows, knees, shins, and feet (Slimani et al. 2017).
Comment 1
Are knee and elbow strikes allowed in all forms of kickboxing?

Line 59-61
Like other CS (such as boxing and MMA), Kickboxing matches are held in weight classes to ensure fairness, and athletes who exceed the specified weight class limit will be disqualified.
Comment 2
If fighters don't lose weight, they can compete in a higher category, which is the case in many CS competitions.

Line 62-63
Past studies have found that most CS athletes lose weight before competition to gain a competitive advantage over their opponents (e.g., fighting an opponent with smaller physicality)
Comment 3
The strength advantage is one reason; another important reason is the rapid weight gain. Fighters who do this are significantly heavier than their opponents in combat.

Line 65-66
Despite evidence that rapid WL can negatively affect athletes' health and performance (PÈlissier 66 et al. 2023; Pettersson et al. 2013; Vasconcelos et al. 2023)
Comment 4
Petterson's main thesis is that those who lose weight have a mental advantage. There are many articles that directly address the health problems associated with weight loss.

Line 91-92
MMA, Muay Thai, and Sambo athletes could stem from (i) a single weigh-in prior to competition; (ii) a weigh-in usually conducted 24-36 hours prior to competition.
Line 103-105
Kickboxing tournaments require weigh-ins on the day of the first match and every morning of the competition days; (iv) if an athlete fails the weigh-in, they are still allowed a second weigh-in opportunity.
Comment 5
Muay Thai is a form of kickboxing, so sentences like this can be confusing. Kickboxing has more forms and more organizations. In the introduction, it must be clear which form of kickboxing this study refers to. In Table 1: Age began training in Sanda (years); Age began competing in Sanda (years). Are the respondents exclusively Sanda fighters?

Line 99-103
The differences between Kickboxing and other CS include: (i) Kickboxing allows attacks using hands, elbows, knees, shins, and feet, but prohibits grappling and ground techniques, which fundamentally distinguishes it from other CS, especially striking-based CS; (ii) the fight will be stopped if a fighter has been knocked down 3 times in the same fight; (iii).
Comment 6
This part is unnecessary, it is not clear how the differences in techniques are related to body weight loss. I think the key differences are in the rules between the various CS, such as: weight tolerance, time from weigh-in to fight, weighing every day of the competition…

Line 108-112
In addition, although previous studies have explored the WL practices of CS athletes, no research has investigated their perceptions of WL, which is a significant gap in the current literature. Understanding athletes' perceptions of WL may help explain why certain harmful practices are adopted despite known risks. Therefore, another aim of this study is to investigate the perceptions of WL among kickboxers.
Comment 7
In your literature, there is a study by Pettersson et al. 2013. The study (Practices of weight regulation among elite athletes in combat sports: a matter of mental advantage?) examines fighters' perceptions of WL. Rephrase this section.

Experimental design

Line 129-148
Chapter Participants
Comment 8
This chapter mainly talks about how the data was collected. Some important data about the sample is missing. Are they competitors in Sanda kickboxing? Most respondents are juniors or teenagers, age 17.2±2.8 (ages between 15 and 17, including 17, are juniors according to Sanda rules).
They have been practicing kickboxing for a little over 3 years and competing for a little over 1 year.
The criterion for inclusion in the study is at least 1 fight in the last year. The group that does not lose weight competes for only 1.5 years. Hypothetically, it is possible that they have only one fight in their career. In these circumstances, the question arises whether we are examining habits or more or less isolated cases.
What is the gender structure of the respondents??

Line 180-185 Chapter Statistical analyses
Participants were categorized by age (junior <18 years old, senior ≥18 years old) competition weight categories (lightweight -51 kg to -63.5 kg, middleweight -67 kg to 182 75 kg, heavyweight -81 kg to -91 kg) (Dugonjić et al. 2019), competitive levels (local, provincial, national, international), attitude toward impact of WL on health (no impact, detrimental to health, beneficial to health), and performance (no impact, detrimental to performance, beneficial to performance).
Comment 9
This part of the text belongs to the chapter participants.

Table 1: Sample Description.
Comment 10
We have a description of the group: Participants who have never engaged in weight loss. There are 48 respondents to whom most of the questionnaire questions do not apply. We do not have descriptions of age groups, weight groups, or competitive level groups.
Comment 11
Did respondents who did not lose weight fill out the questionnaire designed to evaluate rapid weight loss patterns???

Artioli GG, Scagliusi F, Kashiwagura D, Franchini E, Gualano B, and Junior AL. 2010. Development, validity, and 458 reliability of a questionnaire designed to evaluate rapid weight loss patterns in judo players. Scand J Med 459 Sci Sports 20:e177-187. 10.1111/j.1600-0838.2009.00940.

Line 157-158
This study employed the pre-validated RWLQ by Artioli et al. (Artioli et al. 2010), which has been 158 used in previous studies on WL practices in Chinese boxers (Zhong et al. 2024a).
Comment 12
This questionnaire was developed to investigate the phenomenon of rapid weight loss. RWL is the loss of more than 5% of body mass in less than 5 days. In this study, a small number of respondents used RWL. Perhaps another questionnaire on weight loss habits would be more appropriate.

Line 167-168
The content validity of the Chinese version of the questionnaire was evaluated by a researcher with extensive experience in the field of Kickboxing and five Kickboxers through pilot tests.
Line 161-165
The additional questions include the following: (1) the proportion of total BM lost during different stages before competition, (2) who guides the WL process, (3) the weight regain% between weigh-ins and competition, (4) the reasons for WL, (5) the impact of WL on health, (6) the impact of WL on performance, and (7) whether WL leads to unfairness in competitions.
Comment 13
The questionnaire was adapted, and only content validation was performed. Is this version of the questionnaire reliable?

Validity of the findings

The questionnaire to test rapid weight loss habits was conducted on a group of which one-third did not lose weight at all, and most of the others did not lose weight rapidly.

The questionnaire was adapted, and the reliability of the questionnaire was not tested.

The part of the questionnaire that examines perceptions consists of three questions (the effects of WL on health and performance, and whether WL is unfair). This is not enough if you want to investigate fighters' attitudes towards WL more seriously. You have conducted a questionnaire on RWL and are now investigating attitudes to WL? It's not the same, RWL is a brutal version of WL.

Additional comments

Some views of the study could be misleading. e.g., Weight loss practice is fair to other fighters

·

Basic reporting

I have added some comments for your consideration.

L46 – maybe add: ‘relatively low when compared to some other CS’

Intro

L62 – Maybe add a little more here. Practitioners lose weight prior to weigh-in to fall into a particular weight category, then make efforts to gain as much weight between weigh-in and the fight. This is believed to provide a perceived competitive advantage. You could also maybe mention some of the strategies adopted, especially those that are rapid and potentially dangerous (word count permitting)

L92 – There are studies that have looked into this, so it might be worth providing a reference

L124 – Would it not be a purposive sample?

L130 – With the paper questionnaire, verbal instructions were provided, and questions were allowed to be asked. How was this managed for the online participants?

Q154 Do you need to differentiate between rapid weight loss RWL and WL? The scale used is for RWL, but you frequently state you are looking at WL in the Introduction. Does this make a difference? You do report weight loss between 60 – 9 days, so this would not be RWL. It might be worth explaining this somewhere, in relation to the RWL survey questions.

I also think it might be useful to add some sample questions from the RWLQ and include the scoring procedures. For example, is it a Likert Scale?

Beginning of Discussion – might it be worth restating why RWL is used?

L390 - There are studies that have looked into this, so it might be worth providing a reference

Experimental design

No additional comments

Validity of the findings

No additional comments

Additional comments

Overall, very good with some interesting data that does add to our understanding of WL in combat sports. It is well written and is easy to follow. The tables are well-produced and, again, easy to follow. The findings do have important implications for the understanding and prevalence of WL in combat sports. Such information will allow relevant governing bodies to consider how to address this potentially dangerous practice.

The use of a self-selecting, non-probability sample is appropriate in this context. The survey used was a validated instrument, and the additional questions, whilst not validated per se, were piloted. Overall, I think that the instrument allowed for the collection of some very interesting data. Limitations of the design were recognised and highlighted.

Reviewer 3 ·

Basic reporting

The paper “Victory above all: the weight loss practices and perceptions of Chinese kickboxers” aimed to investigate the weight loss practices and perceptions of Chinese Kickboxers. There are a lot of studies that have investigated the weight loss practices in combat sports. However, few studies were conducted involving Chinese kickboxers.

Experimental design

The experimental design is simple but adequate.

Validity of the findings

Some adjustments may be made in the interpretation and discussion of the findings. Specific comments are below.

Additional comments

Abstract
- Change “Kick boxers” and “Kickboxers” to “kickboxers” or “kickboxing athletes”. Review this point in the other sections.
- “The prevalence of WL among Chinese Kickboxers is relatively low.”… Is 68% considered low? Review this assessment here and in the discussion and conclusion.

Introduction
- Change “Kickboxing” to “kickboxing”
- Judo and Sambo must be written in lowercase letters
- The introduction is too long and generic. Please adjust it to be more succinct and direct, making it very clear what is new in this study.

Discussion
- The discussion is also very long and a bit repetitive in some parts; it should be more to the point.
- “This is the first study to investigate the WL practices and perceptions of Chinese Kickboxers.” Be careful with this type of statement. Were systematic searches carried out in all possible databases to verify the existence of other studies?
- Some paragraphs are too long, more than one page long. Adjust this.
- Any explanation for the prevalence of weight loss being lower in kickboxing? In BJJ, for example, one factor that explains it is the weigh-in being almost immediately before the first fight of the competition.
- Was there use of illegal (e.g., diuretics) and/or potentially dangerous methods (e.g., exercising in a sauna, inducing vomiting)? This should be highlighted
- In the introduction or discussion, the impact of weight loss on psychological aspects can be mentioned. E.g., https://doi.org/10.1007/s11332-025-01362-5
- I missed a more objective positioning of how the weight loss process can be conducted based on the data from this study, or how it should be inhibited. The last paragraph of the discussion somewhat addresses this, but I believe it could be more direct.

---

## Round 0.2 · Minor Revisions

· Academic Editor

Minor Revisions

I have mostly added some minor suggestions on improving the readability and clarity in places, so please adjust these aspects where necessary as per below:

- Please check expression of "kickboxers" throughout as sometimes you space them as two separate words (kick boxer) instead of as one for consistency.

- Abstract, results: on the second line change to "being 8.8%" rather than "was 8.8%"

- Abstract, results: change "Coach" to "Coaches" - and also in the conclusions.

- Abstract, results: change "began" to "beginning".

- Abstract, results: remove "div" from the brackets or clarify better.

- Abstract, results: remove the space after the "/" sign in the last sentence.

- Introduction, line 54: should "kickboxing" be capitalised in this title?

- Introduction, line 89: why is a systematic review included to summarise findings when each of the studies are cited above extensively. Could the systematic review possibly be cited on both accounts to sufficiently summarise the scope of the literature and save on excessive citing of individual studies (and space) earlier in this paragraph (and elsewhere if relevant)?

- Introduction, line 100: insert "and" before "(iii)". Do this elsewhere before the final point when giving a numbered list like this in text.

- Methods: I see you have the STROBE checklist cited - which is excellent - but consider incorporating this reporting checklist into the methods to confirm (or add to) the contents reported strictly for surveys - https://pubmed.ncbi.nlm.nih.gov/33886027/.

- Methods, line 182: change "couldn't" to "could not".

- Methods, statistical analyses: Some of the data appear to be categorical rather than continuous based on the survey design (or were they treated as Likert-type ratings in sequence?) - in this way, are Mann-Whitney and Kruskal-Wallis tests able to be used on all the data you collected where applied? If so, please justify the choice briefly within this section.

- Discussion: I agree with reviewer 3 in that some of the content could be condensed in places. Perhaps check further and improve brevity of the explanations/comparisons with past literature throughout this section where possible.

·

Basic reporting

The authors have addressed the issues raised by the reviewers. I think the paper reads well and I commend the authors for their work

Experimental design

The issues raised from the first review have been clarified.

Validity of the findings

no comment

Additional comments

The authors have addressed the comments well, and I think the paper is very good in its current format. Well done for being able to address the issues raised. This is a very important topic, and your work is a good addition to the literature.

Reviewer 3 ·

Basic reporting

The discussion and conclusion sections remain unnecessarily long and, at times, repetitive.

Experimental design

Ok.

Validity of the findings

ok.

Additional comments

Most of the previous concerns have been adequately addressed, and the authors have made relevant modifications in response to the reviewers' comments. However, the discussion and conclusion sections remain unnecessarily long and, at times, repetitive. Given the descriptive nature of the data, a more concise and focused presentation would improve readability and clarity, without compromising the scientific content. The current length is not fully justified by the complexity of the findings.

---

## Round 0.3 · Minor Revisions

· Academic Editor

Minor Revisions

Thank you for making the suggested changes, the manuscript has been strengthened even further. I just have some final minor revisions to make, which are listed below. If you are able to make these changes and return within the next day or two, I can accept this manuscript for publication immediately.

Line 77: Now you have cited the review - thank you for this - perhaps adjust this line to reflect this change such as "Extensive research compiled in a systematic review has investigated...".

Line 85: Change "These studies found" to "This systematic review found" or similar.

Line 274: Remove "both types of" here to simply reflect that coaches, fellow kickboxers, and parents were influential.

Line 314: Change to "in Italy" rather than "in Italian". Why is the country given here but not in other places? Is it because "boxing" is mentioned in multiple places so you wanted to differentiate between countries? If so, MMA and Muay Thai is mentioned on multiple occasions but without identification of country. I would suggest adapting this approach so it is consistent within this section comparing to past research.

Lines 325-327: Move this first sentence to the end of the previous paragraph for direct comparison to MMA here and improved flow. Also, insert a comma after "In contrast" rather than a full-stop on line 325.

Line 330: Perhaps clarify "these athletes" by indicating if you are referring to participants in your study or in past research more clearly.

Line 359: Here you write "five" but in other places throughout the manuscript you give a numeral when referring to number of days, weeks, athletes, etc. So please be consistent in line with what is required for the journal.

Line 444-454: While I appreciate the authors' efforts to make the discussion section more precise, I feel this final paragraph could be removed given it restates the rationale and novelty of the study, which are covered elsewhere. Please consider this suggestion - if you feel this information is essential, perhaps take only pertinent statements from it to include in the conclusion.

---

## Round 0.4 · accepted · Accept

· Academic Editor

Accept

Thank you for addressing my final comments in the previous submission, I am happy to accept your work for publication!